# Allicin Alleviates Diabetes Mellitus by Inhibiting the Formation of Advanced Glycation End Products

**DOI:** 10.3390/molecules27248793

**Published:** 2022-12-12

**Authors:** Linzehao Li, Qinghe Song, Xiandang Zhang, Yan Yan, Xiaolei Wang

**Affiliations:** 1Endocrine and Metabolic Diseases Hospital of Shandong First Medical University, Shandong First Medical University & Shandong Academy of Medical Sciences, Jinan 250001, China; 2Department of Pharmacy, Linyi People’s Hospital, Linyi 276000, China; 3Shandong First Medical University & Shandong Academy of Medical Sciences, Jinan 250118, China; 4Department of Surgery, Qilu Hospital of Shandong University, Jinan 250012, China

**Keywords:** advanced glycation end products, allicin, diabetes, RAGE, oxidative stress

## Abstract

Advanced glycation end products (AGEs) cause damage to pancreatic β-cells and trigger oxidative stress and inflammation, which promotes the development and progression of diabetes and its complications. Therefore, it is important to inhibit the formation of AGEs as part of the treatment of diabetes. Allicin is a natural antimicrobial agent with abundant pharmacological activities, and recent studies have reported its therapeutic effects in diabetes; however, the mechanism of these therapeutic effects is still unclear. Thus, the purpose of this study was to further investigate the association between allicin treatment of diabetes and AGEs. First, we established a streptozocin (STZ)-induced diabetic rat model and treated the rats with allicin for six weeks. We measured glycolipid metabolism, AGE levels, receptor of advanced glycation end products (RAGE) levels, oxidative stress, and other related indicators. The results showed that allicin improved blood glucose and body weight, reduced lipid accumulation, and inhibited AGE formation in rats. Treatment with allicin also inhibited RAGEs and thereby prevented AGE activity, which, in turn, alleviated oxidative stress and promoted insulin secretion. To further verify the effect of allicin on AGEs, we also performed in vitro nonenzymatic glycation simulation experiments. These results showed that allicin inhibited the production of AGEs by suppressing the production of AGEs intermediates. Thus, our research suggests that allicin may alleviate diabetes by inhibiting the formation of AGEs and reducing RAGE levels to relieve oxidative stress and promote insulin secretion.

## 1. Introduction

Advanced glycation end products (AGEs), also known as glycotoxins, are a group of highly heterogeneous and complex harmful compounds produced by nonenzymatic chemical reactions between proteins and reducing sugars or lipids [1,2]. They were first discovered in 1912 by French researcher Louis Camille Maillard, but it was not until the 1960s that researchers identified an abnormal hemoglobin in diabetic patients and named it an AGE. This marked the beginning of research on the association between AGEs, diabetes, and diabetic complications [3,4]. Diabetes mellitus (DM) is a serious chronic metabolic disorder and noncommunicable disease [5]. According to the International Diabetes Federation (IDF), 463 million adults worldwide had diabetes in 2019, and it is predicted that the number of affected people will reach 700 million by 2045 [6,7], highlighting the seriousness of the situation. Studies have shown that AGEs are important contributors to the development of diabetes and its complications [8] and that they induce diabetes by damaging pancreatic β-cells [9]. Further, high levels of AGEs can trigger the production of inflammation and oxidative stress, both of which have a promoting effect on the development of diabetes and its complications [10]. In addition, the hyperglycemic environment of diabetes can, in turn, accelerate the production of AGEs, leading to a vicious cycle.

AGEs can bind to RAGEs to cause oxidative stress and cell proliferation, inducing inflammatory responses in various types of hepatocytes and causing liver lesions and damage to metabolic organs [11,12]. The liver is not only a target of AGEs but also an important site for the clearance and breakdown of AGEs. In 2002, Sebekova et al. first proposed the idea that the metabolism and clearance of circulating AGEs may be impaired by various liver diseases based on the measurement of AGE levels in 51 patients with liver disease and 19 healthy individuals [13]. These data, along with similar findings reported by Yagmur et al. in 2006 [14], indicated that the liver plays an important role in the clearance of circulating AGEs and that liver impairment impacts the ability to clear circulating AGEs. In addition, the liver is an important organ based on its role in glucose metabolism, and impaired liver function caused by AGEs can affect its uptake and utilization of glucose, which is an important factor in the development of diabetes [15]. Therefore, controlling the dietary intake of AGE precursors or inhibiting the production of AGEs is a beneficial measure to prevent diabetes and its complications.

Currently, AGE inhibitors can be classified into three types: AGE formation blockers, AGE lysis agents, and AGE and RAGE receptor binding blockers. For example, aminoguanidine, alagebrium chloride (ALT711), and other inhibitors of AGE formation and cleavage have good effects; however, due to serious side effects and cost of production, none are good candidates for clinical application. In recent years, naturally occurring inhibitors of AGEs have received considerable attention from researchers. In our previous review, we summarized natural compounds that can inhibit the formation of AGEs by scavenging dicarbonyl compounds and antioxidants, lowering the level of blood glucose, and protecting protein structures [16]. These compounds include polyphenols, polysaccharides, flavonoids, and other natural products [17,18,19]. Among them, the use of antioxidant compounds to inhibit the formation and function of AGEs has become a research hotspot in recent years.

Allicin is an active ingredient extracted from garlic, which is a plant of the lily family Alliumceae [20]. Studies have shown that allicin has abundant biologic activities, including antibacterial, antitumor, antioxidant, and diabetic regulatory activities [21,22,23,24,25]. However, there are few reports on whether allicin has inhibitory effects on AGE formation. Therefore, in this study, we investigated the inhibitory effect of allicin on the formation of AGEs through in vivo and in vitro experiments and whether allicin could alleviate diabetes by inhibiting the formation of AGEs.

## 2. Results

### 2.1. Allicin Alleviates Metabolic Dysregulation in Rats with STZ-Induced Diabetes

In this experiment, we first investigated the ability of allicin to improve body weight, blood glucose, and insulin tolerance in diabetic rats. The results from weekly blood glucose monitoring after successful model establishment showed a consistent and significant increase in blood glucose, without intergroup differences. After 6 weeks of treatment with allicin and aminoguanidine, our results showed that the body weight of rats in the model group decreased significantly over time compared with that of the normal control (NC) group. As shown in Figure 1A, the administration of allicin and aminoguanidine led to a slow increase in body weight, indicating that allicin and aminoguanidine had inhibited the decrease in body weight characteristic of diabetic rats. As shown in Figure 1B, after STZ injection, the levels of blood glucose of the rats were all significantly increased. After treatment with allicin and aminoguanidine, the blood glucose of the rats consistently fluctuated within a limited range and was more stable than that of the model group. The allicin group showed a significant decrease in blood glucose in the sixth week. Insulin resistance is one of the main causes of elevated blood glucose and studies have shown that allicin can alleviate insulin resistance. Therefore, before the end of the experiment, insulin resistance tolerance was measured in each group of rats, as shown in Figure 1C. The sensitivity to insulin in the allicin group was significantly greater than that in the model group at each time point from 30 min to 120 min, especially at 30 min and 120 min. This result indicates that allicin may reduce the extent of diabetic insulin resistance.

High-fat diets not only cause elevated serum cholesterol and triglycerides but also provide substrates for formation of AGEs. Therefore, to investigate whether allicin could improve the abnormalities of lipid metabolism in diabetic rats, we analyzed the serum levels of total cholesterol (TC), triglyceride (TG), high-density lipoprotein cholesterol (HDL-C), low-density lipoprotein cholesterol (LDL-C), aspartate aminotransferase (AST), and alanine aminotransferase (ALT) in each group of rats. The results showed that allicin significantly decreased serum TG, TC, and LDL-C levels, as shown in Table 1, while the HDL-C, AST, and ALT levels were not significantly changed (data not shown). In addition, to further observe the lipid-lowering effect of allicin, we performed hematoxylin-eosin (H&E) staining on the liver, as shown in Figure 2. The liver lipid droplets of the model group were large and dense with severe lipid accumulation, while the size and density of droplets decreased significantly after the administration of allicin. In summary, these results suggest that allicin significantly improves the physical and metabolic symptoms in rats with STZ-induced diabetes.

### 2.2. Allicin Reduces AGE Levels in Diabetic Rats

Studies have shown that AGEs can induce diabetes by damaging pancreatic β-cells and that serum levels of AGEs are positively associated with diabetes and its complications. To investigate whether blood glucose and insulin resistance caused by allicin was related to the level of AGEs, we measured the level of AGEs in the serum and liver tissues of diabetic rats. Our results showed that the levels of AGEs in serum increased in both the treatment group and the model group compared with the NC group, with the accumulation of AGEs being more pronounced in the model group. In Figure 3A, after the administration, allicin and aminoguanidine significantly decreased the serum AGE levels compared with the model group. The modulatory effect of aminoguanidine on AGEs was better than that of allicin, although there were not significant differences between the two. The liver is the main metabolic organ of the body and an important site for the catabolism of AGEs. As shown in Figure 3B, the level of AGEs in the liver of rats in the model group was higher compared with those in the NC group, but the difference was not significant. After the administration of allicin and aminoguanidine, the accumulation of liver AGEs in the allicin and aminoguanidine groups decreased significantly compared with the model group.

RAGE is a 42 kDa polypeptide molecule with a unique amino acid terminal sequence and is a multiligand member of the immunoglobulin protein superfamily, which can combine with AGEs to play a physiological role in vivo. Under normal conditions, the expression of RAGE in the organism is low, but in the presence of excess AGEs, RAGE appears at high expression levels. Therefore, to further elucidate the effect of allicin on AGEs, we examined the expression of RAGEs by real-time PCR and Western blotting. The results showed that RAGE gene and protein expression were significantly increased in the model group compared with the NC group. In Figure 3C,D, six weeks after the administration of allicin and aminoguanidine intervention, protein expression decreased significantly in both the allicin and aminoguanidine groups. As shown in Figure 3E, gene expression in the allicin group decreased but did not show significant significance due to intragroup differences. Therefore, our results indicated that allicin prevents AGEs from exerting their biological functions by downregulating RAGEs.

### 2.3. Allicin Alleviates Oxidative Stress in Rats

The accumulation of AGEs increases the formation of reactive oxygen species (ROS) and impairs the antioxidant system. In turn, the formation of AGEs themselves is induced under oxidative conditions, and disruption of the antioxidant system leads to the generation of oxidative stress, which promotes further formation of AGEs. Superoxide dismutase (SOD) is an important indicator reflecting the metabolic status of oxidative free radicals in the human body. SOD play a crucial role in the oxidative and antioxidant balance in the body, and its levels can indirectly reflect the ability of the body to scavenge free radicals. Therefore, we examined and analyzed the oxidation indexes in the serum and liver of rats. The results, shown in Figure 4A,D, indicate both serum and liver SOD levels were significantly decreased compared with those in the NC group. After 6 weeks of allicin and aminoguanidine treatment, the serum and liver SOD levels were significantly increased in the allicin group. The aminoguanidine group showed a significant increase in SOD levels in serum but not in liver tissues, although there was a similar trend. Malondialdehyde (MDA) content is an important parameter reflecting the potential antioxidant capacity of the body. In living organisms, free radicals act on lipids to undergo peroxidation reactions, the end product of which is MDA. MDA is an important intermediate in the formation of AGEs and cross-links, and it polymerizes with proteins, nucleic acids, and other macromolecules. Therefore, measuring the amount of MDA can reflect the degree of lipid peroxidation in the body and indirectly reflect the degree of cellular damage. In Figure 4B,E we show that serum and liver MDA levels were increased in the model group compared with the NC group, especially in liver tissue. The increasing trend of MDA content was apparent, but the trend was not significant due to the high variability of data within the group. Following administration of the treatment, the allicin and aminoguanidine groups showed a slight decrease, but it was not significant. Glutathione peroxidase (GSH-PX) is a very important active enzyme that promotes the degradation of peroxides in vivo and removes the excess residual harmful peroxide metabolites from the cells. In turn, GSH-PX promotes the functional integrity of the organism. As shown in Figure 4C,F, GSH-PX levels were significantly lower in the serum and liver of rats in the model group than those in the NC group. However, both allicin and aminoguanidine were able to elevate the level of GSH-PX to a certain extent, with allicin causing a more significant change. In conclusion, allicin can regulate the levels of SOD, MDA, and GSH-PX, reduce oxidative stress, protect homeostasis of the antioxidant system, and, thus, hinder the formation of AGEs.

### 2.4. Allicin Inhibits the Process of AGE Formation

Animal experiments showed that allicin was able to reduce the level of AGEs. To further verify the ability of allicin to block AGEs formation, we measured the establishment of sugar–protein complexes by UV‒visible absorption spectroscopy. The results in Figure 5 demonstrate that fructose and proteins together were able to generate AGEs. The absorption intensity increase observed at 280 nm may be due to modification of aromatic amino acids, alteration of the microenvironment of amino acid residues and/or formation of nascent chromophores. This result suggests that the aggregation, cross-linking, and unfolding of protein helices during nonenzymatic glycation alters the conformation of bovine albumin (BSA).

An in vitro nonenzymatic glycation system of BSA and fructose was utilized to evaluate the inhibition of protein nonenzymatic glycation by allicin. Fructosamine and α-dicarbonyl compounds are targets used to detect the accumulation of early- and middle-stage glycation products, respectively, which are two important active intermediates in the formation of AGEs. As shown in Figure 6A,B, allicin inhibited the production of fluorescent AGEs in a concentration-dependent manner. At a concentration of 5 mM, allicin significantly inhibited the formation of fluorescent AGEs, with an inhibition rate of approximately 34.78%. This suggests that allicin blocks the cross-linking of proteins and prevents the final stage of AGEs biosynthesis. Allicin has the potential to inhibit fructosamine production, with inhibition rates of 9.39%, 15.90%, and 34.90% when the concentrations of allicin were 0.05, 0.5, and 5 mm, respectively, as shown in Figure 6C. The inhibition rates of aminoguanidine were, correspondingly, 14.58%, 18.27%, and 38.35%, and there was no significant difference in the inhibition rates at the same concentrations. As shown in Figure 6D, allicin inhibited the production of α-dicarbonyl compounds at rates of 16.42%, 22.91%, and 36.03%, respectively. Both these rates and the inhibition rates of aminoguanidine, which were 20.22%, 25.70%, and 42.06%, were dose-dependent. The results of fructosamine and α-dicarbonyl compound experiments showed that allicin could inhibit the formation of AGEs by reducing the oxidation of glycated proteins, inhibiting the formation of Amadori products, scavenging free radicals, and reducing the conversion of fructosamine to α-dicarbonyl compounds.

We used molecular docking techniques to demonstrate the protective effect of allicin on protein glycation sites. The sequence length of bovine serum albumin is 583, and it has two main chains, A and B. We used a molecular docking technique to study the effect of allicin on BSA and the docking results are shown in Figure 7A,B. Allicin has a protective effect on BSA and is surrounded by the polar residue hydrophobic residues, Trp-214, LEU-454, LEU-197, LEU-210, and LEU-480, implying the existence of electrostatic and hydrophobic interactions between allicin and BSA. In addition, allicin formed hydrogen bonds with the Arg-198 residue and occupied arginine sites, which prevented arginine from forming AGEs and contributed to the stability of BSA. This result was similar to the findings of various studies focused on AGE natural medicines. Therefore, it is hypothesized that the binding of allicin to BSA may reduce glycation levels.

## 3. Discussion

AGEs include a large number of structurally and functionally characterized harmful products that damage pancreatic β-cells and, thereby, promote the development of diabetes and its complications [26]. The liver is not only the main site of glucose metabolism but also an important organ for AGE metabolism. Studies have shown that AGEs are closely associated with the further development of liver disease and diabetes. Excessive accumulation of AGEs not only contributes directly to the development of diabetes but also induces inflammation and oxidative damage to liver tissue, leading to a reduction in the liver’s glucose transport capacity [27,28]. Therefore, inhibiting AGEs formation is beneficial for the prevention and treatment of diabetes and its complications. In this study, a natural antimicrobial agent with abundant sources was selected—allicin. Studies have shown that in addition to its antibacterial properties, allicin also has hypolipidemic, antioxidant, anti-inflammatory, and antitumor pharmacological activities [29,30]. Recent studies have shown that allicin has the potential to alleviate diabetes and its complications, but most have focused on its antioxidant and anti-inflammatory effects [31,32]. The current study found that allicin alleviates diabetes by inhibiting the formation of AGEs and reducing RAGE levels to alleviate oxidative stress. This study focuses on the pharmacological activity of allicin and the inhibition of AGE formation and investigates the inhibition of AGE formation by allicin as one of the mechanisms to alleviate diabetes and to provide a starting point for further applications of allicin.

AGEs are not only a cause of diabetes but also a consequence of diabetes [33]. To demonstrate that lowering AGE levels can alleviate diabetes, we first confirmed the effect of allicin on AGE levels through animal experiments. Results of the serum AGE assay showed that allicin was able to reduce serum AGE levels. Next, we analyzed the blood glucose, body weight, and insulin tolerance of the rats. We found that blood glucose in the model group continued to increase over time, while blood glucose in the allicin and aminoguanidine groups was more stable and showed a decreasing trend over time. The results of insulin tolerance test (ITT) experiments also showed that allicin had a mitigating effect on insulin resistance in diabetic patients. In conclusion, our results suggest that allicin can reduce AGEs and blood glucose and ameliorate insulin resistance in diabetic patients, which is consistent with previous studies [34,35].

Diabetes often develops along with hyperlipidemia, and lipid peroxidation products are major sources of precursors for AGE product formation [36]. Lipid peroxidation directly produces AGE precursor compounds, such as reactive dicarbonyl compounds, that cross-link directly with proteins to form stable AGEs [36]. Previous studies have confirmed the hypolipidemic effects of allicin [37,38]. Our results also showed that TG and LDL-C levels were significantly increased in the model group of diabetic rats. TC and HDL-C levels were slightly increased and decreased, respectively, but neither difference was significant. Treatment with allicin resulted in a significant decrease in blood glucose, with the most significant decrease seen for TG and LDL-C. The liver plays a pivotal role in lipid metabolism, lipoprotein synthesis, lipid transport, and as the main site of fatty acid oxidation and ketone body formation [39]. Lipid peroxidation can increase the level of AGEs and lead to liver damage, thus reducing the metabolic function of the liver [26]. H&E staining showed that the liver of model group rats was filled with large and dense fat particles, which were significantly reduced after the administration of allicin treatment. The results of the liver AGE assay showed that the liver AGE level in the model group rats had a tendency to increase compared with that in the NC group, but the difference was not significant. After allicin treatment, the liver AGE level was significantly reduced. Based on the results of serum indexes and serum AGE levels, it appears that the reduction in AGE levels in vivo by allicin may be related to the hypolipidemic activity and the inhibition of lipid peroxidation, which generate AGE active intermediates.

The disruption of the balance between host defenses and antioxidants in the environment could play an important role for development of diabetes and its complications [40], as increased oxidative stress and AGE levels are important factors in triggering diabetes. Oxidative stress is an important part of the body’s metabolic process, often problematic due to the production of excessive ROS and reduced antioxidant capacity, both of which damage the body. Oxidative stress is closely related to hyperglycemia and hyperlipidemia. Persistent hyperglycemia and hyperlipidemia can trigger oxidative stress, mainly via the mitochondrial electron transport chain, sugar autoxidation, and polyol pathways that promote the production of ROS [41]. It has been shown that plasma albumin can be modified by AGEs via increasing oxidative stress, diminishing antioxidant capacity, and impairing hepatic protein hydrolase and respiratory chain enzyme activities [42]. In addition, AGEs bind to RAGEs to promote ROS production, and the level of oxidative stress gradually increases with increasing AGE levels. The levels of SOD, MDA, and GSH-PX were measured in serum and liver tissues, and the results showed that in both serum and liver tissues, SOD and GSH-PX levels were significantly decreased in the model group compared with the NC group, indicating severe oxidative stress in diabetic rats. After allicin and aminoguanidine intervention, we found that SOD and GSH-PX levels were significantly increased compared with the model group. While the results of the aminoguanidine group were significantly different in liver tissues, they were not significantly different in serum, likely due to intragroup variability. Allicin itself showed considerable antioxidant activity, and the regulation of oxidative stress indicators, such as SOD and GSH-PX, would be beneficial to protect the balance between oxidant AGEs and antioxidant capacity in vivo, to reduce the level of oxidation in the organism, and to inhibit the production of AGEs.

Receptor binding is the primary source of AGE activity in vivo, and AGE receptors include RAGEs, scavenger receptors Ⅰ and Ⅱ, oligosaccharyltransferase 48 (OST-48), and galectin-3, among which RAGE is the most thoroughly studied [16]. Under normal conditions, the expression of RAGE is low, but under hyperglycemic and inflammatory conditions, RAGE is highly expressed. AGEs’ binding to RAGEs, which is closely related to vascular and neuropathic conditions, promotes increased ROS levels and affects wound healing in diabetic patients. Our experimental results showed that the level of RAGE expression was significantly increased in the liver tissue of rats in each modeling group compared with the NC group. This finding indicates that the high level of AGEs in the hyperglycemic state leads to the enhancement binding of AGEs and RAGEs because of RAGE overexpression. After intervention with allicin and aminoguanidine, the expression of RAGE was significantly decreased compared with the model group. In conclusion, the combination of hepatic AGE levels and RAGE expression indicated that allicin reduced RAGE expression by inhibiting AGE production.

Finally, the inhibitory effect of allicin on AGEs was verified. Based on the formation process and pathogenic mechanism of AGEs, there are three main methods of AGE inhibition: (1) blocking the formation process of AGEs; (2) degrading the formed AGEs; and (3) blocking the binding of AGEs to RAGEs. Aminoguanidine was the first drug discovered by Prof. A. Cerami in 1986 to be used clinically to inhibit AGE formation, mainly by capturing AGE intermediates to block AGE formation, but it was banned due to severe side effects [43]. ALT711 was the first AGE lysing agent to be used in clinical trials, but the company did not complete the trials due to financial problems. Blocking the binding of AGEs to RAGEs is also a way to inhibit the production of AGEs. This works mainly by blocking the binding of AGEs to RAGEs by using RAGE antibodies to seal them off. However, to date, the most research inhibition method is to block the production of AGE intermediates via antioxidant effects. Therefore, natural compounds with good activity, few side effects, and antioxidant effect have been screened as AGE inhibitors, such as polyphenols, polysaccharides, and flavonoids of natural origin with good antioxidant activity [44,45], all of which are considerable natural AGE formation inhibitors. Allicin is derived from the medicinal and food plant garlic and has biological activities, such as antioxidant, antibacterial, anti-inflammatory, antitumor, and diabetic modulation [30,46]. In this work, to confirm whether allicin has an inhibitory effect on the formation of AGEs, we initially analyzed the inhibitory effect of allicin on the formation of AGEs by in vitro experiments. The results showed that the absorbance of BSA at 280 nm increased in a time-dependent manner when fructose was incubated with BSA at 50 °C for 24–72 h. A new absorption peak at 300–400 nm appeared after 72 h, indicating that the glycation of BSA changed the structure of BSA and produced AGEs. The fluorescence spectra showed that allicin could inhibit the production of fluorescent AGEs and that the inhibition was concentration-dependent. Allicin was able to inhibit fructosamine and α-dicarbonyl compounds, which are intermediate products of AGE formation. Allicin concentrations of 0.05, 0.5, and 5 mm showed concentration-dependent inhibition of fructosamine and active dicarbonyl compounds. In addition, the molecular docking results showed that allicin could bind to the arginine-198 residue on BSA through hydrogen bonding and occupy the arginine site, thus preventing the formation of AGEs by glycation modification of arginine, promoting the stability of the BSA structure, and playing a protein-protective role. These results are similar to the roles of other natural AGE inhibitors in the formation of AGEs. For example, Liu et al. [47] showed that the inhibition of AGE formation by ferulic acid can protect the protein structure via antioxidant activity, scavenging of reactive dicarbonyl compounds, and binding to the corresponding sites of protein arginine or lysine residues. The flavonoids, lapsinoside, baicalin, and naringenin, also have good antioxidant effects and can protect the glycation sites of proteins. In addition, studies on prune polyphenols also suggested that their mechanism of inhibition of AGE formation may be based on scavenging free radicals, trapping AGE intermediate compounds, and reducing protein cross-linking. Allicin is a product with excellent antioxidant activity; therefore, the inhibitory effect of allicin on AGE formation may be related to its own antioxidant capacity. In addition, allicin binds to arginine residue sites with hydrogen bonds, suggesting that it may inhibit the production of AGEs by protecting the susceptible glycation sites.

In this work, we investigated the effects of allicin on the formation of fructosamine, active dicarbonyl compounds, and late AGEs in the process of AGE formation in vivo and in vitro. We have demonstrated that allicin can block the substrate source of AGE formation by regulating blood lipids and oxidative stress in rats. It was concluded that the reduction of AGE level by allicin may be one of the mechanisms to alleviate diabetes and insulin resistance. Although our study has made some progress, more in-depth studies on the mechanisms of allicin inhibition of AGEs formation and alleviation of diabetes in vitro and in vivo are needed at a later stage.

## 4. Materials and Methods

### 4.1. Materials

Allicin (purity ≥ 98%) was purchased from Shanghai Yuanye Biotechnology Co. Ltd. (Shanghai, China). BSA (purity ≥ 98%) was purchased from Sigma Chemical Co. (St. Louis, MO, USA). Fructose and nitro blue tetrazolium (NBT) were purchased from Beijing Solarbio Science & Technology Co., Ltd. (Beijing, China). Aminoguanidine and Girard-T reagent were purchased from Aladdin Chemical Co. (Shanghai, China). An anti-RAGE antibody (ab37647) was purchased from Abcam (Shanghai, China).

### 4.2. Animals

Male SD rats weighing 180–220 g were purchased from the experimental animal center of Shandong First Medical University (Jinan, China). The rats were maintained in our local animal facility in IVC cages at an environmental temperature of 20 ± 1 °C, with a 12 h light/dark cycle and unlimited access to food and water. This study was approved by the Research Ethics Committee of the Shandong Institute of Endocrine and Metabolic Diseases. All animal experiments were conducted in accordance with the Shandong Institute of Endocrine & Metabolic Diseases.

### 4.3. Experimental Design and Treatment

After one week of adaptive feeding, rats were randomly divided into NC (*n* = 10) and STZ-induced groups (*n* = 30). The STZ-induced group was given a high-fat and high-sugar diet for four weeks followed by 12 h of fasting and a secondary intraperitoneal injection of 25 mg/kg of STZ sodium citrate buffer solution (0.1 mol/L, pH 4.4, prepared in an ice bath, and ready for use), and the control rats were injected intraperitoneally with an equal amount of sodium citrate buffer solution. FBG was measured 72 h after injection, and rats were determined to be diabetic models if the mean of two blood glucose measurements was >16.0 mmol/L [48]. At week 7 of the experiment, the STZ-treated group was randomly divided into model group (*n* = 10), aminoguanidine group (60 mg/kg, *n* = 10), and allicin group (40 mg/kg, *n* = 10). Allicin suspension and aminoguanidine solution were prepared with 5% carboxymethylcellulose. The drugs were administered orally, 6 times per week, with a 6-week dosing cycle.

The rats were anesthetized with isoflurane after fasting. They were sacrificed after blood was extracted from the jugular vein. The liver tissues were removed and weighed, and then divided into lyophilized tubes, which were rapidly transferred to −80 ℃ with liquid nitrogen for storage for subsequent experiments.

### 4.4. Fasting Blood Glucose and Body Weight

During the experiment, the blood glucose of the rats was measured by tail vein blood collection using a blood glucose meter ACCU-CHEK (Roche blood glucose meter) at 3 pm every Friday, while body-weight monitoring was performed.

### 4.5. ITT

ITT experiments were performed 6 weeks after administration. Rats fasted for 6 h and blood was collected from the tail vein to determine fasting blood glucose levels by ACCU-CHEK (Roche, Basel, Switzerland). After intraperitoneal injection of 0.75 U/kg insulin, the blood glucose values of the rats were measured at the time points of 0, 30, 60, 90, and 120 min.

### 4.6. Serological Index Measurements

Serological index kits were purchased from TECON Biotech (Ningbo, China). TG, TC, HDL-C, LDL-C, AST, and ALT were measured according to the manufacturer’s directions.

### 4.7. Determination of SOD, MDA, and GSH-PX

The SOD, MDA, and GSH-PX levels were measured using Assay Kits (Nanjing Jiancheng Institute of Biological Engineering, Nanjing, China) according to the manufacturer’s directions, and the concentrations in liver were normalized by the corresponding total protein.

The supernatant was isolated by centrifuging liver homogenates at 4 °C. The serum or liver tissue supernatant was added to the tube with corresponding reagents, vortexed and mixed, and incubated in a warm water at 37 °C for 40 min. The samples were reacted with xanthine (Hydroxylamine method), and the SOD levels were spectrophotometrically measured at 550 nm.

MDA from lipid peroxidation degradation in serum or liver tissue homogenate were condensed with thiobarbituric acid (TBA) to form a red product, and the reaction products were spectrophotometrically measured at 532 nm.

The supernatant was isolated by centrifuging liver homogenates (10%, homogenizing agent: PH 7.4, 0.01 mol/L sucrose, 0.01 mol/L Tris HCl, and 0.0001 mol/L EDTA2Na solution) at 3000 r/min for 10 min at 4 °C. The reduced GSH in the sample were reacted with H_2_O_2_ and formed oxidized GSH. The activity of GSH-PX was expressed as the reaction rate of catalytic GSH subtracted nonenzymatic reactions. GSH and dithiodinitrobenzoic acid interacted to form 5-thiodinitrobenzoic acid anion and was spectrophotometrically measured at 412 nm.

### 4.8. Serum and Liver AGE Determination

A rat AGE enzyme-linked immunoassay (ELISA) kit (Shanghai Enzyme Link Biotechnology Co., Shanghai, China) was used to detect AGE levels in the serum and liver.

### 4.9. Inhibition of AGE Formation by Allicin In Vitro

The preparation of the nonenzymatic glycation system was performed according to the methods of Wang, S., et al. and Zeng, L., et al. [49,50]. Briefly, BSA (20 mg/mL) and fructose (0.5 mol/L) were incubated in PBS (0.2 M, pH = 7.4) in the presence or absence of different concentrations of allicin (0.05, 0.5 and 5 mm) at 50 °C for 24 h. The fluorescence intensity of the mixture was measured at an excitation wavelength of 370 nm and an emission wavelength of 454 nm by an F-320 fluorescence spectrophotometer. Inhibition rate was calculated by the equation: Inhibition rate (%) = ((fluorescence of control − fluorescence of test group)/fluorescence of control) × 100%. The UV absorption spectra in the presence or absence of different concentrations of allicin were measured by UV-2600i spectrophotometer (Shimadzu Experimental Equipment Co., Ltd., Beijing, China). The detection wavelength range was from 240 to 450 nm, and the obtained data were processed with GraphPad 9.0 (Harvey Motulsky, University of California, San Diego, CA, USA).

### 4.10. Determination of the Inhibition Rate of Fructosamine

NBT reagent was used to determine the inhibition rate of fructosamine [51]. Forty microliters of the incubation solution was mixed with 360 μL of ultrapure water and 1.6 mL of 0.3 mM NBT solution (dissolved in 100 mM sodium carbonate buffer, pH = 10.35) at 25 °C for 15 min, and the content of fructosamine was measured by absorption at 530 nm using a Biotek plate reader.

### 4.11. Determination of the Inhibition Rate of α-Dicarbonyl Compounds

Girard-T reagent was used to determine the inhibition rate of α-dicarbonyl compounds [52]. Forty microliters of incubation solution, 160 μL of ultrapure water, and 100 μL of 0.5 M Girard-T reagent (dissolved in ultrapure water) were mixed with 1.7 mL of sodium formate solution (0.5 M, pH = 2.9) and incubated at 25 °C for 1 h. After incubation, the absorbance was measured at 290 nm using a Biotek plate reader (BioTek Instruments, Winooski, VT, USA).

### 4.12. Molecular Docking of Allicin with BSA

A simulation study of the possible binding mode of allicin to BSA was performed using Discovery Studio 4.5 software (Shandong First Medical University, Jinan, China). The crystal structure of bovine serum albumin (PDB ID: 4JK4) was downloaded from the RCSB PDB: Homepage Protein Data Bank, and the structure of allicin was generated with Chem draw 17.0. Prior to molecular docking analysis, allicin (ligand) and BSA (receptor) molecules were pretreated, water molecules were removed from the receptor, hydrogen atoms were added, and the binding site and radius of the receptor were defined. Finally, the ligand was prepared and bound to the receptor.

### 4.13. Staining of Liver Tissue Sections

Tissues of appropriate size were taken from liver of the rats. The tissue blocks were cut into sections and subjected to H&E staining.

### 4.14. Real-Time PCR

In this experiment, total RNA was extracted using the Ultrapure RNA Kit (Beijing Conway Century Biotechnology Co., Ltd., Beijing, China) and was converted to cDNA using the iScript™ cDNA Synthesis Kit. Real-time PCR analysis was used to detect the mRNA expression of RAGE in liver using LightCycler 480 realtime PCR system (Roche, Basel, Switzerland). The primers of RAGE were 5′-ACTCCTCCTTCCAGCTATCGG-3′ and 5′-TTTCCCATCCAAGTGCCAG-3′.

### 4.15. Western Blot

The rat livers were lysed in lysis buffer containing a protease inhibitor. The concentrations of the extracted proteins were determined using the BCA kit (Beyotime Biotechnology, Shanghai, China). The proteins were separated by SDS-PAGE electrophoresis and transferred to PVDF membranes, which were incubated with primary antibody of RAGE at 4 °C overnight. The membranes were subsequently washed with TBST and incubated with the secondary antibodies (Beijing Zhongshan Golden Bridge Biotechnology Co., Beijing, China) for 1 h. Protein detection and densitometric analysis were performed using fluorescence Image System Version 5.2.5 (Odyssey CLx, Cambridge, UK).

### 4.16. Statistical Analysis

All experiments were repeated at least three times. The significance of differences between the groups was estimated using the Student’s *t*-test in GraphPad Prism 9.0. The experimental data are expressed as mean ± standard error of the mean (SEM).

## 5. Conclusions

In this study, the relationship between the hypoglycemic effect of allicin and the inhibition of AGE formation was investigated in a diabetic rat model. We determined serum and liver AGE levels, RAGE expression, oxidative stress, and glucolipid metabolism indexes in rats to show that the inhibition of AGEs by allicin is a potential target of action for the treatment of diabetes. In addition, we demonstrated the inhibitory effects of different concentrations of allicin on AGE formation in the BSA–fructose system by UV and fluorescence spectroscopy techniques and molecular docking, indicating that allicin is an effective inhibitor of AGE formation. The results of this study also provide new insights into the mechanism by which inhibition of AGE formation by allicin alleviates diabetes.

## Figures and Tables

**Figure 1 molecules-27-08793-f001:**
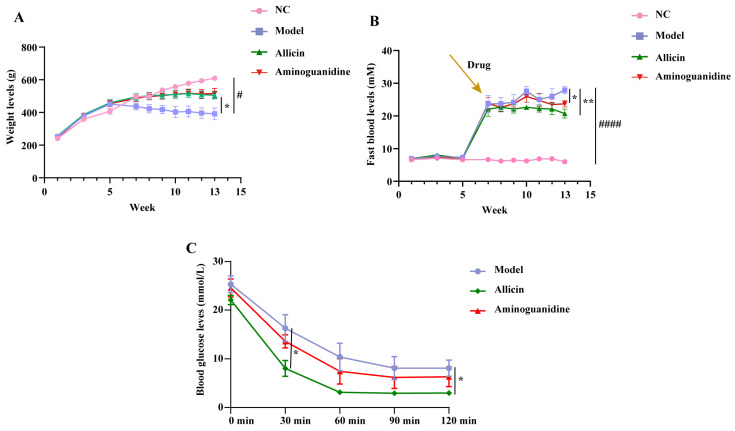
Effect of allicin on the weight and fasting blood glucose (FBG) levels of diabetic rats over the experimental cycle of thirteen weeks. The first week was adaptation feeding, the second week to the fifth week was high-fat induction, and the seventh week was the first week of the drug intervention, which continued for six weeks. The drugs were administered at the end of the 7th week of the experiment, with a 6-week dosing cycle. The experiment ended at the 13th week, and the rats were sacrificed at the same time. (**A**) Change in weight levels during allicin treatment for 6 weeks. (**B**) Change in FBG levels during allicin treatment for 6 weeks. (**C**) At the 12th week, the effects of allicin on the alleviation of insulin resistance in rats after modeling. This assay used aminoguanidine as a positive control. “#” Significant difference between the model group and NC group. “*” Significant difference between the model group and drug group (^####^ *p* < 0.0001, ^#^ *p* < 0.05, ** *p* < 0.01, and * *p* < 0.05).

**Figure 2 molecules-27-08793-f002:**
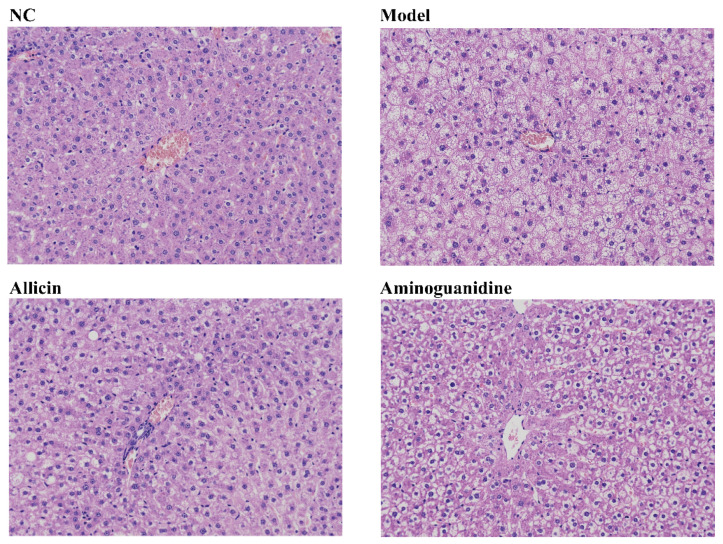
Allicin reduced liver lipid accumulation in diabetic rats. The drugs were administered at the end of the 7th week of the experiment, with a 6-week dosing cycle. The experiment ended at the 13th week, and the rats were sacrificed at the same time.

**Figure 3 molecules-27-08793-f003:**
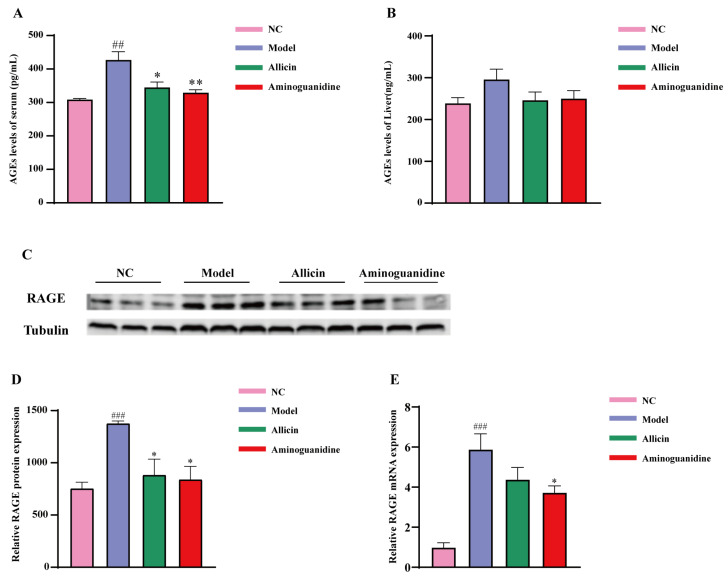
Effects of allicin on AGE levels and RAGE in the serum and liver of diabetic rats. The drugs were administered at the end of the 7th week of the experiment, with a 6-week dosing cycle. The experiment ended at the 13th week, and the rats were sacrificed at the same time. (**A**) AGE levels in serum, (**B**) AGE levels in liver, (**C**,**D**) relative protein expression of RAGE, and (**E**) mRNA expression of RAGE. “#” Significant difference between the model group and NC group. “*” Significant difference between the model group and medicine group. (^###^ *p* < 0.001, ^##^ *p* < 0.01, ** *p* < 0.01, and * *p* < 0.05). The horizontal axes in (A,B,D,E) represent the NC, Model, Allicin and Aminoguanidine group respectively from left to right.

**Figure 4 molecules-27-08793-f004:**
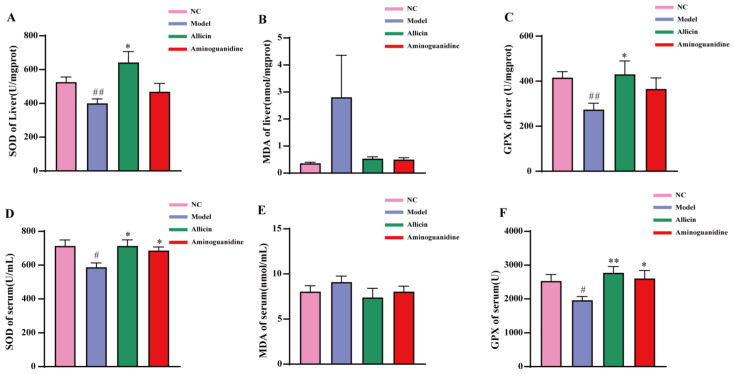
Results of serum and liver SOD, MDA, and GSH-PX measurements. The drugs were administered at the end of the 7th week of the experiment, with a 6-week dosing cycle. The experiment ended at the 13th week, and the rats were sacrificed at the same time. (**A**–**C**) SOD, MDA, and GSH-PX in the liver; (**D**–**F**) SOD, MDA, and GSH-PX in the serum. “#” represents the comparison with the NC; “*” represents the comparison with the model group (^##^ *p* < 0.01, ^#^ *p* < 0.05, ** *p* < 0.01, and * *p* < 0.05). The horizontal axes in Figure 3A, Figure 3B, Figure 3D and Figure 3E represent the NC, Model, Allicin and Aminoguanidine group respectively from left to right.

**Figure 5 molecules-27-08793-f005:**
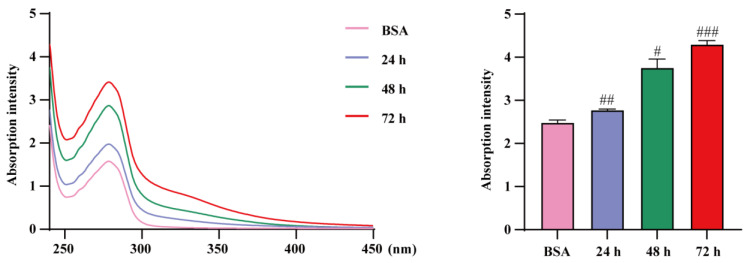
Determination of BSA–fructose UV spectroscopy. “#” represents the comparison with the BSA group (^###^ *p* < 0.001, ^##^ *p* < 0.01, and ^#^ *p* < 0.05).

**Figure 6 molecules-27-08793-f006:**
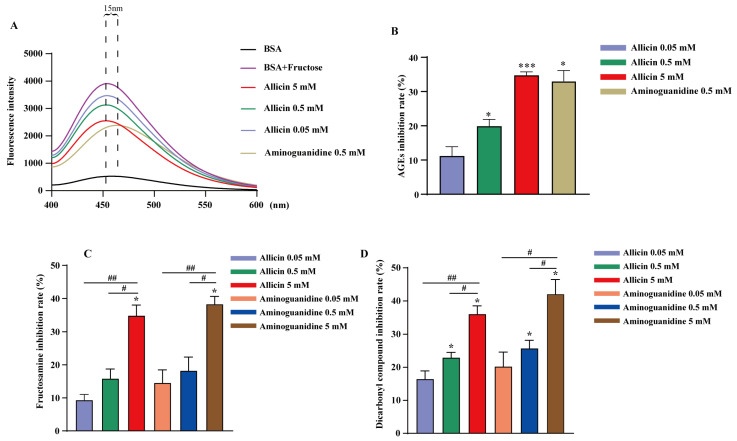
Inhibitory effects of allicin on the production of AGEs in vitro. Fluorescence intensity (**A**), AGE inhibition (**B**), inhibition of fructosamine (**C**), and inhibition of α-dicarbonyl compounds (**D**). The statistical significance of the outcomes was assessed by *t*-tests (*n* = 4). “*” represents statistically significant inhibition of the production of fructosamine, α-dicarbonyl compounds, and fluorescent AGEs by different concentrations of the drug (* *p <* 0.05, *** *p* < 0.001). “#” represents a significant difference in the inhibition production of fructosamine, α-dicarbonyl compounds, and fluorescent AGEs between different concentrations of the same drug (^##^ *p* < 0.01*, ^#^ p <* 0.05).

**Figure 7 molecules-27-08793-f007:**
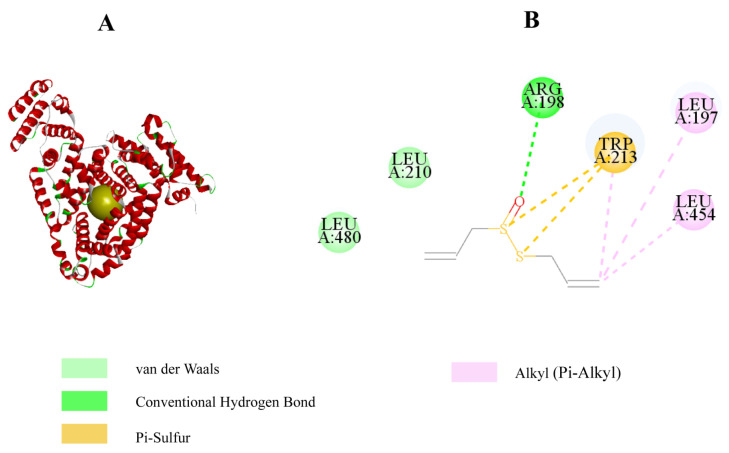
Docking analysis of allicin with BSA by Discovery Studio 4.5. (**A**) Protein Data Bank obtained the molecular structure of BSA (PDB ID: 4JK4), and the best conformation was obtained after docking with allicin by Discovery Studio 4.5 processing. (**B**) Allicin is surrounded by amino acid residues of BSA and interacts with arginine residues of BSA via hydrogen bonding (green dashed line).

**Table 1 molecules-27-08793-t001:** Effect of allicin on blood metabolic indexes in diabetic rats.

Parameters	NC	Model	Allicin	Aminoguanidine
TG (mmol/L)	2.66 ± 0.54	8.52 ± 1.65 ^##^	1.225 ± 0.32 **	1.85 ± 0.59 **
TC (mmol/L)	3.04 ± 0.13	3.42 ± 0.40	2.35 ± 0.19 *	3.03 ± 0.46
HDL-C (mmol/L)	1. 64 ± 0.05	1.48 ± 0.13	1.60 ± 0.15	1.84 ± 0.27
LDL-C (mmol/L)	0.46 ± 0.05	1.00 ± 0.20 ^#^	0.29 ± 0.04 **	0.41 ± 0.09 *

“#” Significant difference between the model group and NC group. “*” Significant difference between the model group and drug group. (^##^ *p* < 0.01, ^#^ *p* < 0.05, ** *p* < 0.01, and * *p* < 0.05).

## Data Availability

Data are available from the authors.

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
