# Peer review of "Allicin Alleviates Diabetes Mellitus by Inhibiting the Formation of Advanced Glycation End Products"

_molecules, 2022, doi:10.3390/molecules27248793_

Round 1
Reviewer 1 Report
In general, numerous typographical and grammatical errors have been observed. The English, formulation of certain sentences, and paragraphs need great improvement to provide readers better understandability of the text.
Materials & methods:
Essential information to describe methodology should be included: e.g., animal ethics clearance statement, AG and allicin route and frequency of administration, vehicle for AG and allicin, method of euthanasia, tissues collected post-euthanasia.
Additionally, the authors need to further elaborate methodology in these sections to be sufficiently detailed: 2.4 (fasting blood glucose determined using glucometer?), 2.7 (via ELISA?)
Section 2.9 and 2.12 should be combined.
Histology of liver, qPCR and western blotting of RAGE were not included in the methodology section.
Results
Please double check the statistical analysis for Fig 3B: NC vs Model not significant, but significant when Model vs Allicin and AG.
Fig 3C: Was densitometer used to scan the blot (since the authors claimed RAGE significantly decreased in allicin and AG groups)?
Discussion
Emphasize the limitation, new and important aspects of the current study.
Author Response
Thank you for these valuable feedbacks, we have studied the comments carefully and made corrections in the manuscript according to your comments.
In general, numerous typographical and grammatical errors have been observed. The English, formulation of certain sentences, and paragraphs need great improvement to provide readers better understandability of the text.
Response: Thank you for your kindly suggestion. We have revised English grammatical errors and reformatted the paper in the order of “Introduction, Results, Discussion, Methods, and Conclusion”.
Materials & methods:
Essential information to describe methodology should be included: e.g., animal ethics clearance statement, AG and allicin route and frequency of administration, vehicle for AG and allicin, method of euthanasia, tissues collected post-euthanasia.
Response: Thank you for your comments.We have improved the information about materials and methods. We added animal ethical clearance statement in paragraph 4.2, lines 4-5. Based on your suggestion, in paragraph 4.3, we have added the route and frequency of administration of allicin, vehicle of allicin and aminoguanidine, the method of euthanasia and the associated tissues retained.
Additionally, the authors need to further elaborate methodology in these sections to be sufficiently detailed: 2.4 (fasting blood glucose determined using glucometer?), 2.7 (via ELISA?)
Response: Thank you. After reformatting, paragraph 2.4 becomes paragraph 4.4, paragraph 2.7 becomes paragraph 4.7. In paragraph 4,4, we used the Roche blood glucose meter ACCU-CHEK for fasting glucose measurements. In paragraph 4.7, we used the kit from Nanjing Jiancheng Biotechnology Company for the determination of SOD, MDA, GSH-PX and the methods were added in the methods section.
Section 2.9 and 2.12 should be combined.
Histology of liver, qPCR and western blotting of RAGE were not included in the methodology section.
Response: Thank you for pointing this out. Following your suggestion, we have merged paragraphs 2.9 and 2.12. After reformatting, paragraph 2.9 becomes paragraph 4.9. In addition, we have added histology of liver, qPCR and western blotting of RAGE in the methodology section.
Results
Please double check the statistical analysis for Fig 3B: NC vs Model not significant, but significant when Model vs Allicin and AG.
Response: Thank you, we apologize for our mistake. There is no significant difference between the groups in Fig 3B. We have completely rechecked the statistical analysis throughout the paper and there was no more mistake.
Fig 3C: Was densitometer used to scan the blot (since the authors claimed RAGE significantly decreased in allicin and AG groups)?
Response: Thank you for your suggestion. We have used densitometer to scan the blots and show the result as Figure3D.
Discussion
Emphasize the limitation, new and important aspects of the current study.
Response: Thank you for pointing this out. We added a paragraph at the end of the Discussion section.
Reviewer 2 Report
The manuscript entitled '' Allicin alleviates diabetes mellitus by inhibiting the formation of advanced glycation end products'' provide new, scientifically important data about effects of influence of allicin on formation of advanced glycation end products (AGE) in vivo on STZ-induced diabetic rat model and in an adequate in vitro model. Moreover, the author investigated the influence of allicin consumption on lipidemia, liver function, parameter of redox status and oxidative damage of lipids-MDA, as all these factors could be involved in allicin-induced reduction of AGE presence.
The authors used appropriate routine methods for determination of parameters of interest, but used methods are not described in Material and methods section including method for H&E staining on the liver, real-time PCR and Western blotting for PAGE determination, method for preparation of lipid extract for determination of antioxidant enzymes and MDA. Moreover, I suggest to the authors to carefully rewrite section about antioxidant enzymes and MDA determination, to add some description or/and weather they used commercially available kits or/and some reference. To improve the quality of material and methods section they should also add the names for producer of ELISA kit for rat AGE. Moreover, the authors stated that they determined ALT and AST by they do not show results for these parameters and further discuss obtained results.
Some other minor suggestions and comments
The authors should carefully check if all abbreviation are described when they are used for the first time in Introduction section (AGE, IDF, RAGE...).
I suggest to authors not to cite references in results section. The descriptions that stands in results sections are correct, but all these facts are general, well known and there is not need for references citation in results section.
For example. High-fat diets not only cause elevated serum cholesterol and triglycerides but also provide substrates for the formation of AGEs (32). Could be modified to High-fat diets not only cause elevated serum cholesterol and triglycerides but also provide substrates for formation of AGEs. (The citation could be deleted.)
The rewriting of the sentence in discussion section, the first paragraph starting with Studies have shown.. should be considered as it is too long and it is not clear.
The references are missing after sentence in discussion section, the first paragraph Recent studies have shown that allicin has the potential to alleviate diabetes....
In discussion section, the first sentence could be carefully rewritten as lipid peroxidation products are major sources of precursors for AGE product formation.
In discussion section, the first sentence could be modified to The disruption of the balance between host defenses and antioxidants in the environment could play an important role for development of diabetes and its complications...or similar
Hyperglycemia and hyperlipidemia or increased glucose or lipid levels can be used instead of hyperlipidemia and hyperlipidemia levels in discussion section.
There is a spacing mistake in conclusion section it write effectiveinhibitor and a spelling mistake in section 2.9. It stands...concentration of allicin (0.05, 0.5 and 5 mm) instead of allicin (0.05, 0.5, 5 mM).
The manuscript could be published in current form, but the more detail description of methods could add scientific value and improve overall the quality of the paper, by my opinion.
Author Response
Thank you for these valuable feedbacks, we have studied the comments carefully and made corrections in the manuscript according to your comments.
The authors used appropriate routine methods for determination of parameters of interest, but used methods are not described in Material and methods section including method for H&E staining on the liver, real-time PCR and Western blotting for RAGE determination, method for preparation of lipid extract for determination of antioxidant enzymes and MDA. Moreover, I suggest to the authors to carefully rewrite section about antioxidant enzymes and MDA determination, to add some description or/and weather they used commercially available kits or/and some reference. To improve the quality of material and methods section they should also add the names for producer of ELISA kit for rat AGE. Moreover, the authors stated that they determined ALT and AST by they do not show results for these parameters and further discuss obtained results.
Response: Thank you for your comments.
- We have added the method for liver H&E staining, real-time PCR and western blot in the methods and materials section (see paragraphs 4.13-4.15).
- We measured SOD, MDA, GSH-PX levels using kits from Nanjing Jiancheng Biotechnology Co. and the methods were added in the methods section, please see 4.7.
- We added the name of the ELISA kit manufacturer (Shanghai Enzyme Link Biotechnology Co.) in paragraph 4.8.
- We apologize for our mistake. Since the serum AST and ALT levels were not significantly changed, so the data were not shown.
Some other minor suggestions and comments
The authors should carefully check if all abbreviation are described when they are used for the first time in Introduction section (AGE, IDF, RAGE...).
Response: Thank you for pointing this out. We have described in detail all the abbreviations used for the first time in the introduction section and corrected the errors in the full text.
I suggest to authors not to cite references in results section. The descriptions that stands in results sections are correct, but all these facts are general, well known and there is not need for references citation in results section.
Response: Thank you for your suggestion. We have removed some references cited in the results section and and have reorderd the list.
For example. High-fat diets not only cause elevated serum cholesterol and triglycerides but also provide substrates for the formation of AGEs (32). Could be modified to High-fat diets not only cause elevated serum cholesterol and triglycerides but also provide substrates for formation of AGEs. (The citation could be deleted.)
Response: We agree with you. We have removed the reference.
The rewriting of the sentence in discussion section, the first paragraph starting with Studies have shown.. should be considered as it is too long and it is not clear.
Response: We apologize for our mistake. We have rewritten this sentence, see lines 4-8 of the first paragraph of the discussion section.Thank you for your comments.
The references are missing after sentence in discussion section, the first paragraph Recent studies have shown that allicin has the potential to alleviate diabetes....
Response: Thank you for pointing this out. We have checked and added references.
In discussion section, the first sentence could be carefully rewritten as lipid peroxidation products are major sources of precursors for AGE product formation.
Response: We agree with you. We have changed the “lipid peroxidation is one of the major sources of AGE production” to “lipid peroxidation products are major sources of precursors for AGE product formation” according to your suggestions. Thank you for your suggestion.
In discussion section, the first sentence could be modified to The disruption of the balance between host defenses and antioxidants in the environment could play an important role for development of diabetes and its complications...or similar
Response: Thank you for your suggestion. We have changed the “A potential explanation for the prevalence of diabetes is disruption of the balance between host defenses and oxidants (AGEs) in the environment” to “The disruption of the balance between host defenses and antioxidants in the environment could play an important role for development of diabetes and its complications”.
Hyperglycemia and hyperlipidemia or increased glucose or lipid levels can be used instead of hyperlipidemia and hyperlipidemia levels in discussion section.
Response: Thank you for giving the replacement option. We agree with you. Indeed, “hyperglycemia or hyperlipidemia levels” is a wrong description. The “hyperglycemia or hyperlipidemia levels” have been replaced by “hyperglycemia or hyperlipidemia” throughout the manuscripts.
There is a spacing mistake in conclusion section it write effectiveinhibitor and a spelling mistake in section 2.9. It stands...concentration of allicin (0.05, 0.5 and 5 mm) instead of allicin (0.05, 0.5, 5 mM).
Response: Thank you for pointing this out. The spacing mistake in conclusion section has been corrected. The concentration of allicin “0.05, 0.5 and 5 mM” has been replaced by “0.05, 0.5 and 5 mm” throughout the manuscripts.
The manuscript could be published in current form, but the more detail description of methods could add scientific value and improve overall the quality of the paper, by my opinion.
Response: Thank you for your suggestion. We read the full manuscript and carefully revised the details and methods. Thank you again.
Round 2
Reviewer 1 Report
The authors have improved the manuscript but there are still numerous typographical errors in the manuscript.
Author Response
Thank you for these valuable feedbacks, we have studied the comments carefully and made corrections in the manuscript according to your comments.
- Figure 1C, there is no unit for the time. Need to specify the unit(days or hours).
Response: Thank you for your kindly suggestion. We have added time units (minutes) in Figure 1C, and enriched the information in legends of Figure 1. Now, the revised Figure 1 is more rigorous and the information is clearer
- No detailed information was provided in all the figures' legends. For example, Figure 2 shows that allicin reduced liver lipid accumulation in diabetic rats, but at what time points? One can dig into the text and find it out, but providing the details in the legends helps readers interpret the figures and tables.
Response: Thank you. The drugs were administered at the end of the 7th week of the experiment with a 6-week dosing cycle. The experiment ended at the 13th week and the rats were sacrificed at the same time.We now add time points in the legends of Figures 1 to 4, so that the readers can get information more clearly in the legends.
- Text polishing is still needed. For example, On page 5, it reads "RAGE is a 42 kDa polypeptide molecule..." But the next sentence, it says that "RAGE is the most important pathway...". Is RAGE a polypeptide or a pathway?
Response: Thank you for pointing this out. We have modified this imprecise statement and optimized some other language problems, especially in page 2, 7, 12, etc. You can see them in the article. We really appreciate your valuable time and meticulous work. Thank you again.